

# Fuzzy information recognition and translation processing in English interpretation based on a generalized maximum likelihood ratio algorithm

Li Yin

Chengdu Gingko College of Hospitality Management, Chengdu, Sichuan, China

## ABSTRACT

English interpretation plays a vital role as a critical link in cross-language communication. However, there are various types of ambiguous information in many interpreting scenarios, such as ambiguity, ambiguous vocabulary, and syntactic structures, which may lead to inaccuracies and fluency issues in translation. This article proposes a method based on the generalized maximum likelihood ratio algorithm (GLR) to identify and process fuzzy information in English interpretation to improve the quality and efficiency of performance. Firstly, we systematically analyzed the common types of fuzzy information in interpretation and delved into the basic principles and applications of the generalized maximum likelihood ratio algorithm. This algorithm is widely used in natural language processing to solve uncertainty problems and has robust modeling and inference capabilities, making it suitable for handling fuzzy information in interpretation. Then, we propose a fuzzy information recognition model based on the generalized maximum likelihood ratio algorithm. This model utilizes a large-scale interpretation corpus for training and identifies potential fuzzy information in the interpretation process through statistical analysis and pattern recognition. Once fuzzy information is detected, we adopt a series of effective translation processing strategies, including contextual inference and adaptation, to ensure the accuracy and naturalness of interpretation. Finally, we conducted a series of experiments to evaluate the performance of the proposed method. The experimental results show that the fuzzy information recognition and translation processing method based on the generalized maximum likelihood ratio algorithm performs well in different interpretation scenarios, significantly improving the quality and fluency of interpretation and reducing ambiguity caused by fuzzy information.

Corresponding author
Li Yin, li.yin@gingkoc.edu.cn

# INTRODUCTION

Language constitutes an indispensable facet of human daily communication, serving as a primary medium for information exchange and interaction. In escalating globalization, the seamless conversion of diverse languages assumes paramount significance. Language

conversion is accomplished in formal settings through interpretation, demanding precision, standardization, and celerity (*Crowston, Allen & Heckman, 2012*; *Gupta, Jain & Joshi, 2018*; *Carvalho Joao, Batista & Coheur, 2012*). However, amidst the emergence and application of language, vagueness pervades.

Ambiguity is an inherent aspect of human language, denoting the uncertainty that envelops word scope. As cognition is subject to human arbitrariness and subjectivity, fuzzy language invariably emerges, representing a fundamental attribute of human thought (*Deepa, Vani & Singh, 2014*; *Wang et al., 2023*). The human brain notably possesses the capacity to recognize and process external information in a fuzzy manner. Some scholars once posited that semantic fuzziness fundamentally mirrors the uncertainty surrounding the categorization and essence of concepts within human comprehension, thus emanating from language's role as the material embodiment of thought (*Ying, Tan & Zhang, 2018*). Fuzzy information includes language elements with ambiguity, ambiguity, uncertainty, or ambiguity in language or text, which makes the interpretation and understanding of information uncertain or difficult. Vague information may lead to different interpretations and understandings, so special care needs to be taken in communication, translation, and interpretation processes to ensure accurate information communication.

Crucially, concepts serve as the bedrock of linguistic meaning elements, though semantics and concepts diverge in categorization. Concepts reside within the domain of thought, while semantics pertains to language's sphere. Human thought capacity is profoundly developed, yet linguistic expression of concepts remains comparatively constrained (*Bobillo et al., 2009*; *Yu, Yuan & Wu, 2023*). Consequently, language must employ the fewest linguistic units to convey the utmost information, often necessitating the utilization of the same word to express different concepts. Hence, certain language words and grammatical elements inevitably harbor semantic ambiguity (*Gao & Yang, 2020*).

Computer technology exhibits remarkable potential in the realm of interpretation, and intelligent information technology has revolutionized people's lives (*Wan et al., 2023*). This impact becomes even more significant in intensifying globalization, where effective communication between diverse languages is paramount. Moreover, the prevalence of instant messaging is an inexorable trend in modern society. The study of English interpretation is a profoundly technical endeavor, necessitating the synergistic collaboration of multiple disciplines. With computer technology backing, it can furnish more precise and prompt interpretation outcomes (*Dan & Srivastava, 2023*; *Tomsovic et al., 1987*). This study will focus on applying vague language in oral communication, covering situations from different social and professional fields. We will analyze various types of vague language, including euphemisms, vague vocabulary, speculative expressions, and their practical applications in communication.

To attain elevated precision in English interpretation, the GLR algorithm is harnessed to recognize speech data information (*Wang, 2021*; *Lin et al., 2021*; *Duan, 2021*). A syntactic function based on an analytical, linear table is derived by rectifying structural ambiguity in Chinese and English. The primary contributions of this article are as follows:

(1) The method devised in this article proficiently conducts an automatic search for phrases.
(2) The method proposed herein adeptly identifies various instances of fuzzy information.

(3) The method put forth in this article enhances the accuracy of machine interpretation.

The main content of this article is arranged as follows. Section 'Related Work' shows the related work. Section 'Research Principle of Intelligent Recognition Language Technology' introduces the research principle of intelligent recognition language technology. The information fusion between different interpretation languages is presented in 'Information Fusion Between Different Interpretation Languages'. The research on fuzzy information processing in English interpretation based on an improved GLR algorithm is shown in 'Research on Fuzzy Information Processing in English Interpretation Based on Improved GLR Algorithm'. Section 'Experimental Results' shows the experimental results and the conclusions are described in 'Conclusions'.

This study has the following practical significance. Firstly, vague language is a common phenomenon in oral communication; in some situations, it is a necessary component of effective communication. Secondly, understanding and applying vague language can improve communication effectiveness and reduce misunderstandings and conflicts. Thirdly, we live in a diverse society where people from different cultures and backgrounds may use vague language in different ways and frequencies. Therefore, studying vague language can help promote understanding and success in cross-cultural communication.

Therefore, solving the problem of vague language in oral communication is crucial for improving communication quality, accuracy, and efficiency. This can be achieved by improving the clarity of language expression, providing clear information, and using vague language to balance accuracy and flexibility. This will help improve personal and organizational communication skills, promote better interpersonal relationships, and create a more efficient work environment. Portions of this text were previously published as part of a preprint (*Yin, 2023*).

## RELATED WORK

Machine interpretation is one of the research contents in the field of NLP. There are several methods, as follows.

Rule-based approach. At first, the linguistic school used this method to analyze the source language text by context-sensitive grammar, then translate it, and finally output the voice by computer. In theory, this method has the highest accuracy. However, the formal grammar of the natural language is highly complex, difficult to summarize, and large in number. The rule-based method is highly complex, has high requirements for the correctness of rules, and the accuracy is easy to degrade (*Anandika, Sujata & BijayKumar, 2023*; *Andrade Sequoia & Hannah, 2023*).

Statistical-based approach. After research, the complexity of translation methods based on statistical methods is relatively low. Compared with the rule-based method, the translation based on statistics is smoother. For a long time, the statistics-based method has been the traditional method of machine interpretation (*Dimmy, Lima & Pozo, 2023*).

Recurrent neural networks (RNNs)-based approach. With the development of deep learning technology, the method based on neural networks has become one of the hot fields. RNNs have been used for machine translation, and long-short-term memory (LSTM),

gate recurrent unit (GRU) and other structures have emerged. With the emergence and development of multi-core processors, the shortcomings of RNNs that cannot be parallelized are exposed. To overcome this problem, relevant scholars have proposed a coder–decoder model, which can be combined with an attention mechanism, residual connection, and other technologies. In 2017, Google proposed to use the self-attention transformer model to deal with machine translation problems, abandoning the practice of using RNN/convolutional neural networks (CNN) and only using the attention mechanism (*Jantscher et al., 2023*; *Song, Li & Wareewanich, 2023*).

For Natural language processing, rule-based Machine translation (RBMT), corpus-based interpretation, and machine learning are effective methods for solving interpretation problems (*Marzieh, Ghovatmand & Nazemi, 2020*). Machine translation is a technology that uses computer programs to translate words or sounds from one natural language to another. However, this method has limited ability to handle fuzzy information.

In machine intelligence recognition algorithms, the phrase part of speech recognition is particularly important, as it can handle many grammatical ambiguities in phrases, sentences, and words. Using annotated phrase corpus content, words in short sentences can be divided. The words in English sentences exist independently, and the segmentation processing of Chinese words and sentences is implemented to determine the translated sentence and word part of speech. Finally, syntax is used to analyze the dependency relationship of phrases and create a sentence syntax tree. The GLR algorithm is widely used in speech recognition to determine the relationship between the preceding and following phrases. It uses dynamic recognition forms as the basis for unconditional transfer statements. GLR did not detect grammatical ambiguity while translating phrases, requiring re-duplication and calibration. If grammatical ambiguity is detected, the parsing linear table is linearly called using syntactic analysis geometry to identify the content of the phrase. The principle of local optimization is used to improve the quality of the content, and symbols are placed through different recognition channels to improve the accuracy of the recognition results.

# RESEARCH PRINCIPLE OF INTELLIGENT RECOGNITION LANGUAGE TECHNOLOGY

Research principles should include the following aspects: First, when language signals are sent out, they will be arranged according to the time sequence, and then the language information will be encoded and converted into a coding form that the computer can recognize. Secondly, after the language information is correctly encoded according to the time sequence, this encoding can transmit the content with acoustic signals and express different coding languages with different discrete symbols. Finally, through the intelligent perception of the computer, the specific semantics and voice of the language can be distinguished. The language's scene structure, grammar and semantics are expressed in a form similar to the human voice using computer intelligence recognition technology (*Al-Absi Abdullah, Hassan & Bashir Shaban, 2011*; *Oscar, Quirin & Sánchez, 2008*).

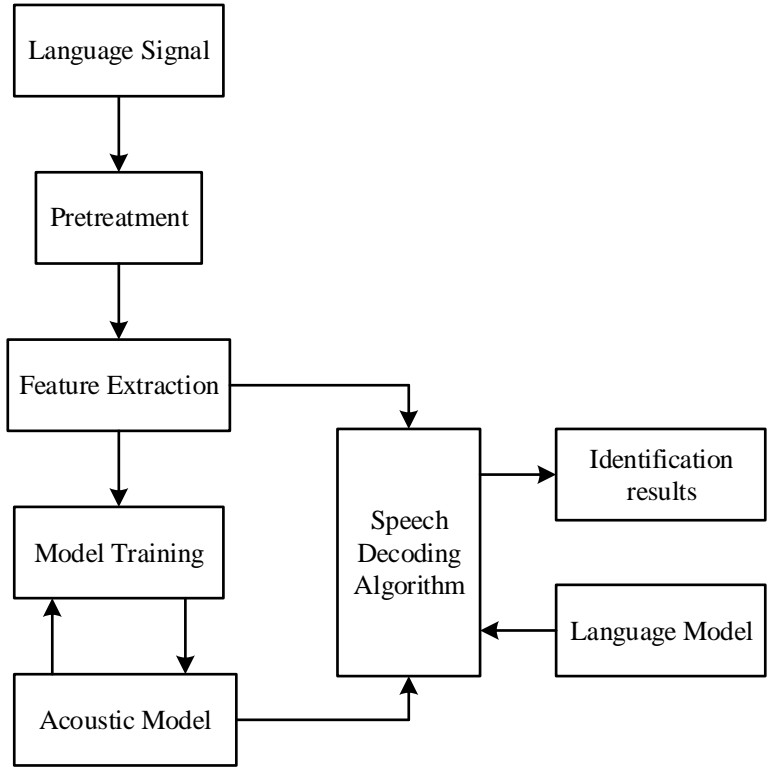

**Figure 1**  **Structure block diagram of language recognition based on statistical pattern.**

The language model is mainly constructed in the way of the statistical model. This is because the semantics and voice of language have similar frequencies to a certain extent. Modeling in statistical mode will improve the accuracy of interpretation. The structure block diagram of language recognition based on statistical method is shown in Fig. 1.

The identification method consists of the following parts.

## Feature extraction

The feature extraction of speech signal processing needs to analyze the speech signal. First, it needs to analyze and extract the feature parameters that can represent the essence of the speech. Only with feature parameters can these feature parameters be used for effective processing. Different from parameter extraction, speech signal analysis can be divided into time domain, frequency domain and other domain analysis methods. The speech signal analysis method can be divided into model and non-model analysis methods. The template mainly extracts relevant features from the acquired language information for acoustic model processing. At the same time, in the whole feature extraction, we should also pay attention to the influence of external interference and noise (*Olivetti et al., 2020*).

## Statistical acoustic module

During voice transmission, various obstacles of diverse sizes, shapes, and characteristics populate the acoustic environment, engendering an exceedingly intricate soundscape. To

safeguard against distortions in the transmission process, the statistical acoustic module typically employs a first-order hidden Markov model. This model is instrumental in preserving concordance between the acoustic module's emitted sound and the received signal. In the current phase of intelligent speech recognition research, language analysis is grounded in N-ary grammar within the realm of statistics.

### Decoder module

The Decoder module is one of the core parts of intelligent NLP. Its main task is to output the correctly recognized signal string with the maximum probability through statistical acoustics, linguistics, and other disciplines.

The working principle of the decoder is as follows:

First, the input language's language signal or feature extraction is obtained according to intelligent recognition technology. Now, $m_t$ is used to represent the voice vector sequence under the condition of time, which can be specifically expressed as:

$$M = \{m_1, m_2, \cdots, m_L\} \tag{1}$$

Then, intelligent language recognition technology can be expressed as:

$$W^* = \underbrace{\mathrm{argmax}}_{w}\{P(W,M)\} = \underbrace{\mathrm{argmax}}_{w}\left\{\frac{P(WM)}{P(M)}\right\} \tag{2}$$

where $W = \{w_1, w_2, \cdots, w_L\}$ is a sequence of words, which is a continuous language.

Through Bayesian rule, Eq. (2) can be written as

$$= \underbrace{\mathrm{argmax}}_{w}\left\{\frac{P(M|W)P(W)}{P(M)}\right\} = \underbrace{\mathrm{argmax}}_{w}(P(M|W)P(W)) \tag{3}$$

Since the input language string is specific $m$, its $P(M)$ is also determined. Therefore, omitting it will not affect the final result.

$$W^* = \underbrace{\mathrm{argmax}}_{w}(P(W|M)) = (PW)$$

If in the intelligent recognition system, the words $w_i$ and $w_j$ are independent of each other, then their input values $m_i$ and $m_j$ are also independent of each other and $m_i$ is highly sensitive to $w_i$ (*Tomsovic et al., 1987*), then the other form of Eq. (4) is

$$= \underbrace{\mathrm{argmax}}_{w_1, w_2, \cdots, w_L}\left\{\sum_{i=1}^{L}\log(P(m_i|w_i)) + \log(P(w_i))\right\} \tag{4}$$

where $P(m_i|w_i)$ is the number of occurrences of $w_i$ in the text data. Use the text data corresponding to the voice database to directly calculate $P(w_i)$, namely.

$$P(w_i) = \frac{P(m_i|w_i)}{H} \tag{5}$$

where $H$ represents the total number of occurrences of all words in the text data.

## Principles of computer translation

Semantic scrutiny of the source terms constitutes an imperative prerequisite for machine translation. A correlation between the source statement and its corresponding English counterpart can be established by adhering to the directives governing the conversion of Chinese to English grammar. This process concurrently generates the translated output expression, culminating the machine translation process (*ElHossainy Tarek, Zeyada & Abdelkawy, 2023*; *Zhumadillayeva et al., 2020*).

The following formula can express the translation.

$$NE_x = \frac{\omega \times S \times H_{vv}}{k^* G_{xy}} \eta \tag{6}$$

where $\omega$ is the extracted source statement characteristics; $S$ is any entry in the dictionary; $k^*$ is the value of entries in different grammatical formats; $\eta$ is the knowledge redundancy.

# INFORMATION FUSION BETWEEN DIFFERENT INTERPRETATION LANGUAGES

## Fusion function

Assuming that there are $N$ different kinds of interpretation language information, $m$ features can be extracted from each language information, and there are $K$ words of the same type in each feature, then the membership function can be expressed as

$$\mu = \begin{bmatrix} \mu_{11}(x_1) & \cdots & \mu_{1K}(x_1) \\ \vdots & \ddots & \vdots \\ \mu_{N1}(x_N) & \cdots & \mu_{NK}(x_N) \end{bmatrix} \tag{7}$$

where $u_{NK}(x_N)$ is the membership function of the $N-$ type words under a particular feature of the Kth interpretation language.

It can be seen that if the two interpretation languages are mutually supportive, the difference between their membership functions is relatively small. Therefore, the degree of mutual support between different interpretation languages can be expressed by the difference between membership functions. In this article, European distance is introduced to express this difference.

$$d_{ij} = \sqrt{\left(\mu_i - \mu_j\right)\left(\mu_i - \mu_j\right)^T} \tag{8}$$

where $\mu_i$ and $\mu_j$ are line vectors. Its physical meaning is that a certain feature of the $i$th interpretation language and the $j$th interpretation language belongs to the membership vector of all words. One result can be obtained:

$$d_{ij} = \sqrt{\sum_{k=1}^{K} \left(u_{ik}(x_i) - u_{jk}(x_j)\right)^2} \tag{9}$$

where $0 \leq d_{ij} \leq 1$ is the confidence distance, the greater the $d_{ij}$, the lower the level of support between the two interpretation languages, and the higher the contrary. Then, the fusion function of the two interpretation languages is defined as

$$h_{ij} = 1 - d_{ij} \tag{10}$$

Therefore, if the $i$ and $j$ interpretation languages can be expressed as fusion functions, then the fusion matrix can be expressed as

$$h = \begin{bmatrix} h_{11} & \cdots & h_{1N} \\ \vdots & \ddots & \vdots \\ h_{N1} & \cdots & h_{NN} \end{bmatrix} \tag{11}$$

Now, given a fusion vector $g = [g_1, g_2, \cdots, g_N]^T$. To represent the interpretation language that can be recognized by all other interpretation languages, to ensure maximum reliability, it is necessary to determine the minimum amount of fusion between other interpretation languages and the $i$-th interpretation language, $i.e.,$

$$g_i = \min(h_{i1}, h_{i2}, \cdots, h_{iN}) \tag{12}$$

where $g_i$ indicates the degree of integration between the ith interpretation language and other interpretation languages, and bringing Eqs. (10) into (11) will get the following results.

$$g_i = \min\left(1 - \sqrt{\sum_{k=1}^{K} \left(\mu_{ik}(x_i) - \mu_{jk}(x_j)\right)^2}\right), j = 1, 2, \cdots, N \tag{13}$$

## Information fusion between different interpretation languages

In this article, it is necessary to describe the same features between different interpretation languages. The weight coefficient can express the proportion of different interpretation languages in the integration

$$l = [l_1, l_2, \cdots, l_N]^T. \tag{14}$$

As mentioned earlier, the larger the $g_i$, the higher the degree of recognition of the $i$-th interpretation language by other interpretation languages, and the greater the fusion weight. Therefore, a hypothesis is given as follows.

$$l_i = \frac{g_i}{\sum_{i=1}^{N} g_i} \tag{15}$$

where $\sum_{i=1}^{N} l_i = 1$.

Then, the membership vector for different interpretation languages is

$$\alpha = \eta^T \times l = [\alpha_1, \alpha_2, \cdots, \alpha_K] \tag{16}$$

where $\alpha_K$ represents the membership of the fused k-type words and

$$\alpha_K = \sum_{i=1}^{N} (\mu_{ik}(x_i) l_i). \tag{17}$$

The method described above can complete the integration of different interpretation languages. The same method can also be used for feature fusion for different features.

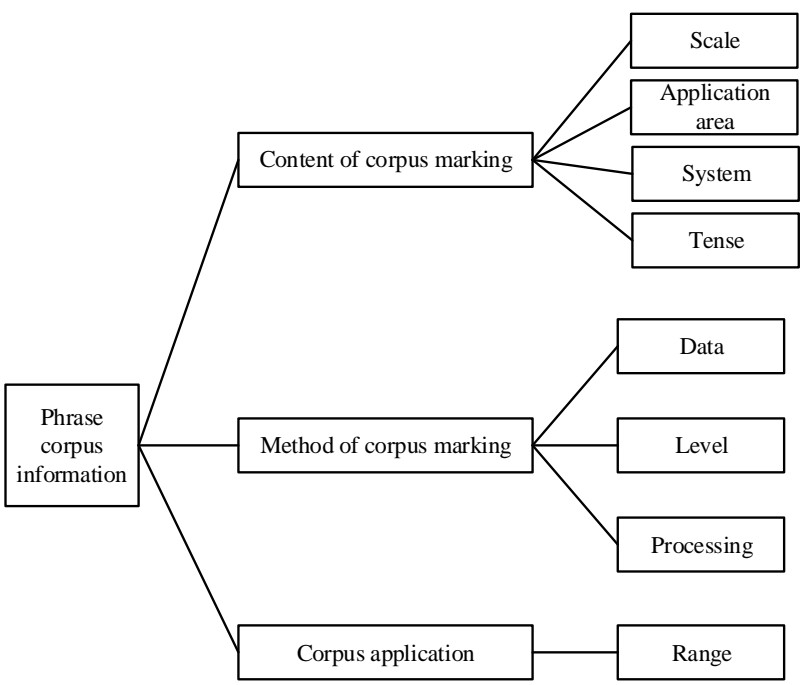

**Figure 2** Structure block diagram of language recognition based on statistical pattern.

Information fusion can improve the coverage and quality of interpretation, especially in multilingual and cross-cultural communication. However, it also faces challenges such as handling multilingual ambiguity, ensuring consistency and accuracy of information, *etc*. In the next chapter, we will use an improved GLR method to handle ambiguity and ambiguity between different pieces of information.

# RESEARCH ON FUZZY INFORMATION PROCESSING IN ENGLISH INTERPRETATION BASED ON IMPROVED GLR ALGORITHM

## Constructing phrase corpus

In English interpretation, the establishment of a corpus is essential. In the process of interpretation, it can accurately match the parts of speech of English and Chinese phrases and effectively improve the accuracy of machine interpretation. The information flow of the phrase library is shown in Fig. 2. This article marks the English–Chinese phrase corpus and distinguishes the tenses of different phrase corpora.

Phrase speech recognition is the foundational component of intelligent interpretation, encompassing the ability to process a diverse range of words and effectively handle unclear sentence structures (*Giordano et al., 2022*). The significance of this aspect lies in its capacity to accurately decipher and interpret spoken language, thereby forming a fundamental pillar of the entire interpretation process.

**Table 1  The classification of hedges in vague English.** By categorizing and understanding the various types of hedges employed in English, the interpretation process can be further optimized, ensuring more precise and contextually appropriate translations.

| English | |
| --- | --- |
| Category | Example |
| Modal auxiliary | can, may, |
| Modal verbs | seem, guess |
| Modal adjective | possible, likely |
| Adverbial of manner | almost, perhaps |
| Modal noun | possibility, probability |

Furthermore, the unique nature of English sentences, wherein each word exists independently, facilitates dividing and categorizing text into individual words and sentences. This characteristic feature enables a more granular and precise analysis of linguistic elements, contributing to the overall comprehension and interpretation of the language. This research employs syntax analysis to construct a comprehensive syntax tree to bolster the interpretation accuracy. By hierarchically structuring the language, the proposed methods enhance the ability to capture the underlying grammatical relationships within the text, resulting in more refined and accurate interpretations. As depicted in Table 1, the article presents a classification of hedges in vague English. These hedges are linguistic devices that express uncertainty or vagueness in communication. By categorizing and understanding the various types of hedges employed in English, the interpretation process can be further optimized, ensuring more precise and contextually appropriate translations.

## Improved GLR method

In the field of spoken English, structural ambiguity is one of the complex problems. This study introduces an intelligent English interpretation model *via* an improved GLR method.

GLR method can interpret the relationship between sentences and translate them. If the semantic ambiguity is not detected during the translation process, the GLR algorithm must be calibrated again. Local optimization improves the content quality if ambiguity is seen in the statement. Different identification channels are used to identify symbols and improve the accuracy of identification results.

Generally, the traditional GLR algorithm is entirely accidental and has a significant coincidence probability in identifying data, which cannot meet the current accuracy. The accuracy of machine interpretation can be enhanced by the improved GLR  method, which is shown in Eq. (18):

$$G_E = (V_N, V_T, S, \alpha) \tag{18}$$

where $S$ is the start symbol cluster; $V_N$ is the circular symbol cluster; $V_T$ is the termination symbol cluster; $\alpha$ is phrase action cluster.

## Correction methods of English interpretation

In the traditional English interpretation problem, its final result can be obtained by speech recognition. However, part of speech recognition does not change the ambiguity

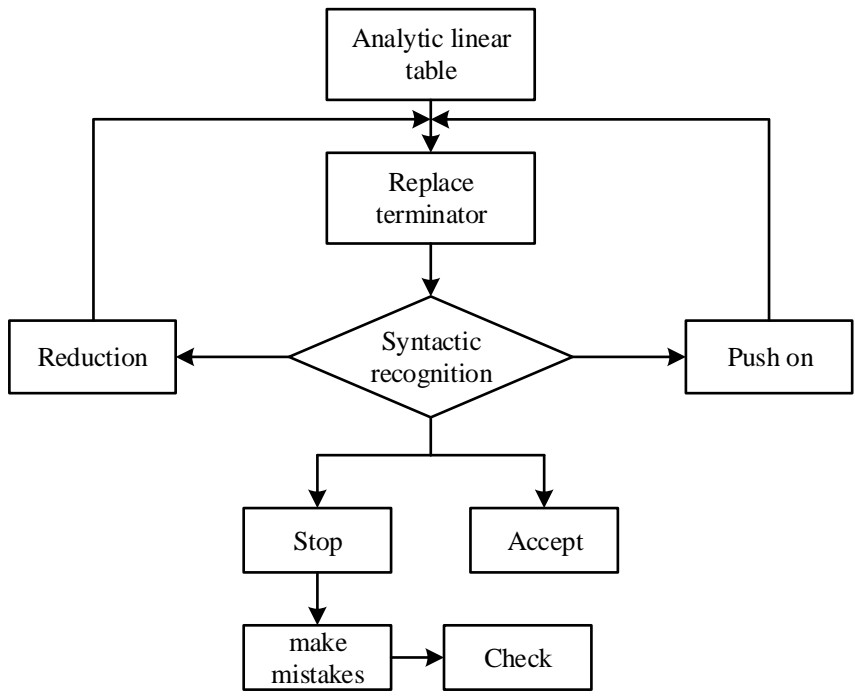

**Figure 3  Algorithm correction flow chart.**

between structures. Therefore, it is necessary to correct structural ambiguity in English interpretation. The analytic linear table is used for phrase recognition, which examines the areas of error in the findings of part-of-speech detection. With the pointers of advance, reduction, acceptance, termination and error, correct the error points by searching the phrase marker content in the corpus. Figure 3 shows the algorithm correction flow chart. Multiple recognition results can be written into the same node of the phrase structure tree when multiple recognition results must be output simultaneously (for instance, when two phrases are adjacent in the sentence). The receiver pointer will then consider it automatically as one recognition result.

## A step-by-step method for Fuzzy language processing based on improved GLR technology

Improving GLR technology is an advanced method for handling fuzzy languages. The following is a step-by-step method for improved GLR technology.

**Data preprocessing:** Preprocessing input text or voice data. This includes word segmentation, removal of stop words, stem extraction, *etc.*, to reduce noise and extract key information.

**Feature extraction:** Extracting features from text or speech data that aid in recognizing fuzzy information. Features can include part of speech tags, emotional scores, syntactic structures, *etc.*

**Fuzzy information recognition:** In improved GLR technology, the recognition of fuzzy information is a key step. The combination of GLR algorithm and enhancement technology

can identify and label fuzzy information in text or speech data, such as euphemisms, uncertain expressions, emotional meanings, *etc*.

**Explanation generation:** Improved GLR technology generates corresponding explanations or translations once fuzzy information is recognized. This can involve contextual inference, emotional interpretation, grammar correction, *etc*., to ensure the accuracy and naturalness of the explanation.

**Performance evaluation:** Finally, evaluate the performance of the improved GLR technology. This includes comparing its performance on different tasks and datasets to ensure the effectiveness and accuracy of fuzzy language processing.

# EXPERIMENTAL RESULTS

## Experimental scheme

The impact of the proposed method was tested on the processing of fuzzy information in interpretation. The interpretation evaluation mainly consists of the following aspects: interpretation accuracy, interpretation speed and updating ability. Professional interpreters and computers completed the experimental group.

Test process.

Dataset selection. A dataset containing 50 voice files with fuzzy semantics was used to test the model's performance. These voice files can contain various types of ambiguous information, such as polysemy, metaphors, speculative expressions, *etc*., to simulate situations encountered in actual interpretation tasks.

Test conditions. The experiment is divided into two test conditions. Manual interpretation: Under this condition, professional interpreters translate voice files. Interpreters use their interpretation skills to interpret the content of voice files based on their understanding and experience.

Machine interpretation: Under this condition, use the proposed method for machine interpretation. This includes converting voice files into text and using models to process fuzzy information.

Evaluation indicators.

The indicators for evaluating interpreting performance include.

Interpretation accuracy: Compare the interpretation results with the original voice file's content to evaluate the interpretation accuracy. It can be measured using indicators such as accuracy, F1, or BLEU scores.

Interpretation speed: Evaluate the time an interpreter or machine takes to complete the interpretation. It can be measured in words per minute or speech duration per minute.

Update ability: Evaluate the ability of interpreters or machines to process ambiguous information, including handling polysemy, speculative expressions, and understanding metaphors.

The details of these experimental settings help to ensure the rigor and comparability of the experiment while also providing a means to evaluate the proposed method's performance comprehensively. Evaluating multiple indicators such as interpretation accuracy, speed, and updating ability can help researchers comprehensively understand the method's effectiveness and draw accurate conclusions. The scoring rules are shown in Table 2.

**Table 2  Scoring rule table.** The rater will score different algorithms through comparison. The scoring rules are as shown.

| Project | Scoring rules |
|---|---|
| Identification accuracy | Whether the evaluation content is clear and the structure is reasonable. |
| Identification speed | Total recognition time divided by the number of recognition phrases |
| Update capability | Total recognition update time divided by the number of recognition phrases |

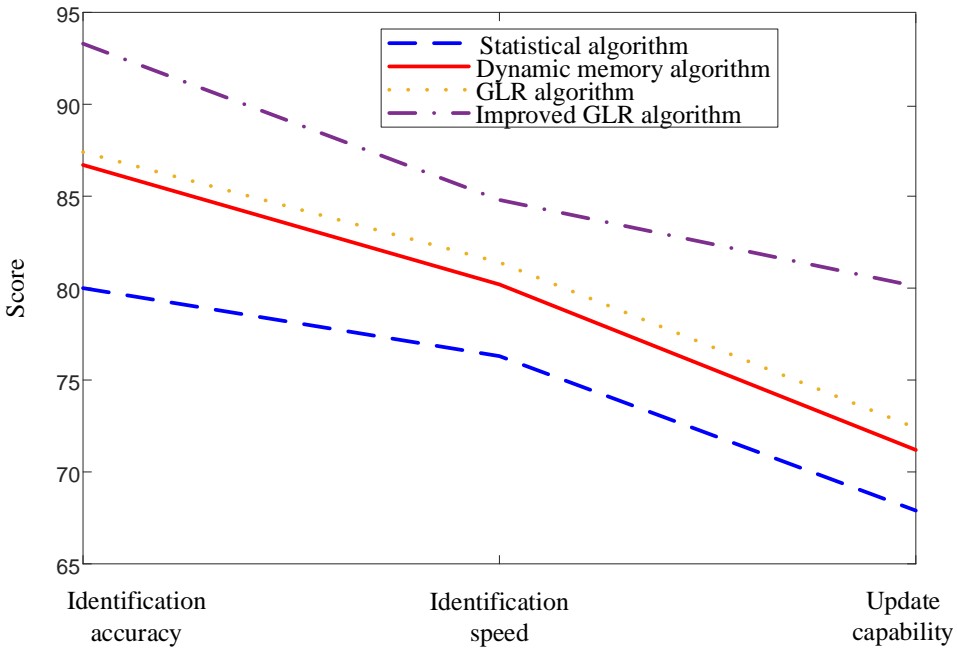

**Figure 4  Comparison of interpretation results of four methods.** The figure presents a clear comparison of the proposed model against other existing models, highlighting its exceptional performance in terms of recognition accuracy, recognition speed, and updating ability for part-of-speech translation.

## Comparison of scores of different methods

Figure 4 presents a clear comparison of the proposed model against other existing models, highlighting its exceptional performance in recognition accuracy, recognition speed, and updating ability for part-of-speech translation. The results unequivocally demonstrate the superiority of the proposed model over its counterparts.

A practical case study is conducted to ensure a comprehensive evaluation involving the interpretation of actual statements using statistical methods, dynamic memory methods, and the GLR algorithm. The proposed algorithm's and other algorithms' performance are meticulously compared, and the results are depicted in Fig. 5. In this illustrative example, the English statements requiring interpretation are as follows: "Mary has married with what's his name–you know, the man with the curly blond hair," "We will give you our reply

as soon as possible," and "The joint venture established by the two sides was somewhat of a failure".

Figure 5 shows that only the interpretation results generated by the proposed model exhibit the closest resemblance to human interpretation. The recognition accuracy rate surpasses 96%, reaching levels akin to professional interpreters. This remarkable accuracy further underscores the superiority of the proposed algorithm. The success of the proposed model can be attributed to its advanced techniques and innovative approach to language interpretation. Leveraging statistical methods, dynamic memory methods, and the GLR algorithm, the model showcases its adeptness in comprehending and accurately interpreting the nuances present in the given statements. It is worth mentioning that achieving such a high level of accuracy and proximity to human interpretation marks a significant milestone in Natural Language Processing. As demonstrated in the example sentences, the proposed algorithm's ability to tackle vague and intricate English information holds immense potential for real-world applications in various domains.

The experimental results provide impressive evidence that fully supports the effectiveness of the proposed method. The analysis in Figs. 4 and 5 demonstrates the superior performance of the proposed model compared to other existing models, especially in recognition accuracy, recognition speed, and interpretation accuracy.

Firstly, the comparison in Fig. 4 indicates that the proposed model exhibits excellent accuracy in fuzzy information recognition. The significant improvement in recognition accuracy indicates that this method can handle various types of fuzzy information effectively. In addition, the recognition speed of the model is also a key indicator, and experimental results show that the proposed method is competitive in processing speed, which is crucial for real-time interpretation tasks in practical applications.

Secondly, the actual case study in Fig. 5 further emphasizes the excellent performance of the proposed model. When interpreting actual English statements, the model achieved recognition accuracy of over 96%, approaching the level of human interpreters. This demonstrates the model's ability to handle fuzzy information and highlights its feasibility and accuracy in practical applications.

Most importantly, these experimental results provide support for the proposed method and mark an important milestone in the language processing field. The achievement of such high accuracy and interpretive ability close to the human level has further promoted the development of natural language processing technology, bringing new possibilities for various language-related applications and fields. This indicates that the proposed method has broad application prospects and is expected to be important in improving interpretation and language comprehension tasks.

In summary, the experimental results strongly support the proposed method's effectiveness and highlight its potential and prospects in the language processing field. These results provide a solid foundation for further research and application and are expected to promote the development of interpretation and language understanding fields.

To thoroughly assess the algorithm's effectiveness across diverse scenarios, this research employs the proposed algorithm and other established algorithms to conduct a meticulous comparative experiment using the Susanne English tree database. The

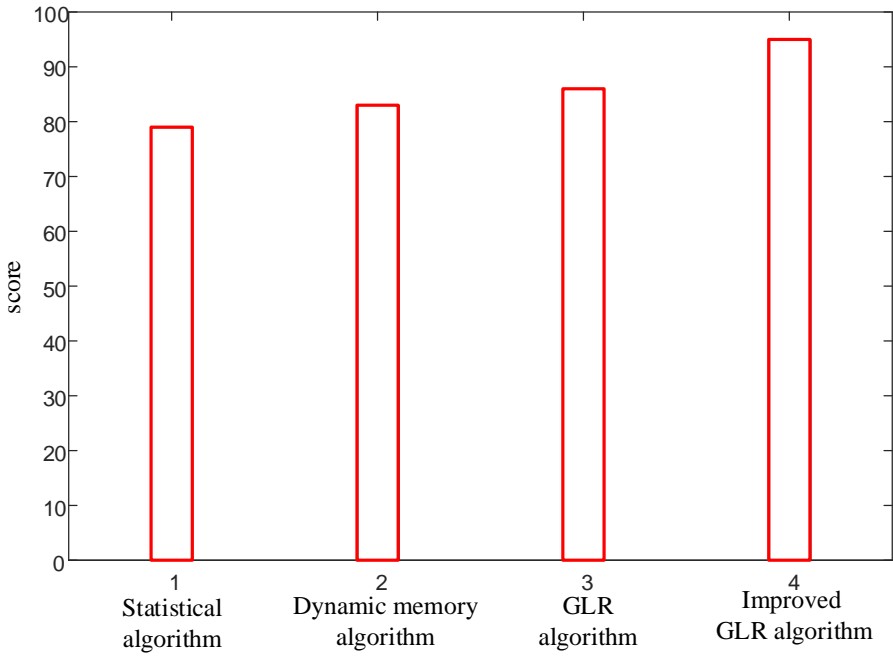

**Figure 5** **Comparison of different methods.** In conclusion, the experimental findings firmly establish the superiority of the proposed algorithm over alternative models.

database encompasses 64 files, containing a comprehensive corpus of 150,000 words. We meticulously derived 457 production expressions from this corpus, facilitating the generation of 40 context-free ambiguous grammars.

The experimental procedure commenced with the rigorous training of a distinct network model for each grammar, ensuring the models captured the intricacies of the respective grammatical structures. Subsequently, we crafted 50 unique sentences for each grammar, culminating in a diverse dataset of 2,000 sentences subjected to thorough analysis utilizing diverse algorithms. The experimental findings, eloquently presented in Fig. 6, conclusively indicate that the algorithm proposed in this article exhibits superior accuracy in handling grammatical complexities within general English databases. These promising results underscore the algorithm's potential to significantly enhance NLP applications and foster a deeper understanding of language structures.

However, it is imperative to acknowledge the limitations of this study. Although the Susanne database is invaluable, representing key aspects of general English language patterns, its coverage may not fully encapsulate the vast linguistic nuances in real-world language usage. As such, future investigations should encompass broader and more diverse datasets to evaluate the algorithm's generalizability and robustness comprehensively.

For instance, under the method that does not differentiate the degree of abstraction based on parts of speech, expressions such as "arise smoke" and "attack pirate" are erroneously categorized as abstract concepts, even mistakenly regarded as fuzzy expressions. In contrast, our proposed method accurately identifies them as concrete concepts, leading to correct

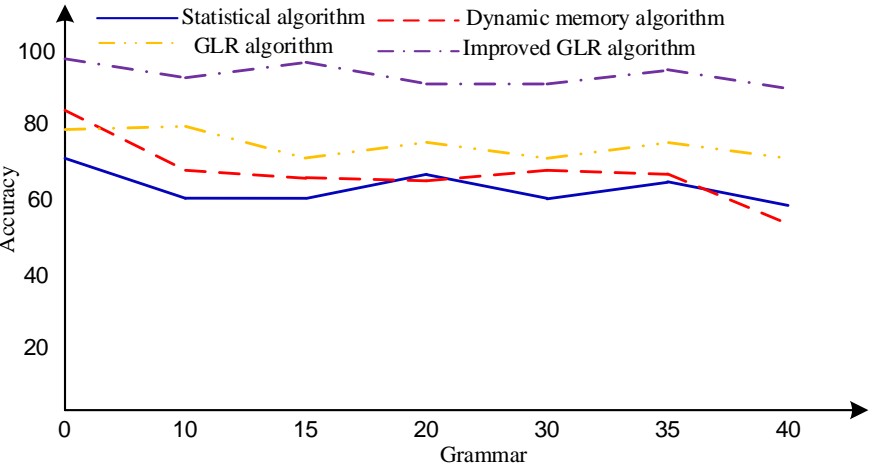

**Figure 6 Accuracy of sentence analysis corresponding to 40 grammars.** For instance, under the method that does not differentiate the degree of abstraction based on parts of speech, expressions such as "arise smoke" and "attack pirate" are erroneously categorized as abstract concepts, even mistakenly regarded as fuzzy expressions.

recognition outcomes. Words like "smoke" and "pirate" possess various parts of speech, each exhibiting significant disparities in their fuzzy characteristics.

Additionally, the conventional model mistakenly interprets the images of "coldwoman" and "woman" as synonymous due to their high similarity, overlooking the metaphorical nature of "coldwoman." Conversely, our model effectively captures the implicit fuzzy information within the metaphorical expression, successfully discerning its underlying meaning. While "woman" is a concrete word, our model employs its modal features for precise fuzzy recognition.

In summary, this research successfully verifies the efficacy of the proposed algorithm in various contexts through a meticulous comparative analysis using the Susanne English tree database. The attained outcomes not only contribute to the advancement of NLP research but also hold the potential to elevate the quality of language-related applications, paving the way for further algorithmic refinements and advancements in the field.

## Identification node distribution comparison

To fully verify the advantages of the proposed method in this study, the English interpretation proofreading test of the model in the text is realized through experiments, the data in the experiment process is recorded, and the system's performance is analyzed. In the experiment, there are 400 characters to proofread the vocabulary, 500 short articles to proofread the vocabulary, and 25 kb/s word recognition speed. Compare the accuracy before and after proofreading; the results are shown in Table 3.

Table 1 shows the highest accuracy rate before and after proofreading is 75.2% and 99.6%, respectively. In addition, the average accuracy rate before and after proofreading is 68.52% and 98.72%, respectively. Due to the distinct differences in accuracy between

**Table 3  Compare of the accuracy.** Comparison of the accuracy before and after proofreading.

| Experiment number | Before proofreading (%) | After proofreading (%) |
|---|---|---|
| 1 | 56.4 | 98.5 |
| 2 | 69.5 | 99.2 |
| 3 | 70.2 | 98.2 |
| 4 | 75.2 | 99.6 |
| 5 | 71.3 | 98.1 |
| Average value | 68.52 | 98.72 |

the two methods, it is demonstrated that the intelligent recognition model of systematic English interpretation is effective.

## Discussion

The rapid advancements in computer and artificial intelligence technology have led to significant breakthroughs in interpretation technology, particularly in fuzzy information interpretation. This article presents a novel fuzzy information interpretation method based on the improved GLR (generalized LR) algorithm, which bears immense practical significance:

(1) Enhanced interpretation accuracy: The fuzzy information interpretation technology excels in handling challenging scenarios, such as instances with strong accents, rapid speech, and unclear pronunciation. By leveraging its robust capabilities, the technology comprehends and processes voice information more effectively, elevating the accuracy of interpretation. This improvement proves to be of great value in diverse contexts, ensuring more precise and reliable communication.

(2) Streamlining meetings and negotiations: In critical meetings and negotiations, various environmental factors, including ambient noise and cross-talk, may interfere with the clarity of information. The fuzzy information computer interpretation technology acts as an invaluable asset by adeptly discerning and understanding speech content amidst such adversities. Consequently, this technology significantly enhances the efficiency and outcomes of meetings and negotiations, enabling participants to comprehend complex dialogues easily and quickly.

(3) Supporting special groups: This technology's significance is its ability to assist individuals with language or hearing impairments. By employing fuzzy information computer interpretation technology, these individuals can better grasp voice information, leading to more accessible and convenient communication methods tailored to their unique needs. This inclusive approach fosters greater inclusivity and communication for all society members, reinforcing the proposed method's societal impact and humanitarian value.

In conclusion, developing the fuzzy information interpretation method based on the improved GLR algorithm marks a milestone in interpretation technology. Its advantages in enhancing interpretation accuracy, streamlining meetings and negotiations, and supporting

special groups underscore its practical importance. This innovative technology addresses real-world challenges and paves the way for more inclusive and efficient communication in an increasingly diverse and interconnected world. As such, it can potentially revolutionize various sectors and significantly enrich human-machine interactions in the future.

## CONCLUSIONS

Compared to translation, the process of interpretation adeptly manages a spectrum of ambiguous information with swifter and more refined precision. This article introduces an enhanced GLR methodology meticulously tailored to address the intricacies of fuzzy information encountered in computer interpretation. In contrast to conventional techniques, the proposed approach achieves heightened precision and adaptability. Computerized translation facilitates English learners in navigating essential dialogues encompassing intricate verses and nebulous concepts. The integration of fuzzy semantics within communication and writing processes seamlessly enhances sentence flow and coherence, amplifying the conveyed content's expressive impact. Only through diligent inquiry and exploration of these challenges can the domain of computer interpretation make significant strides. Anchored in a professional fuzzy semantic framework, this study enhances translation accuracy. It empowers readers to discern emotional nuances, with an aspiration to contribute to translating and communicating literary masterpieces.

### Funding
The author received no specific funding for this study.

### Competing Interests
The authors declare there are no competing interests.

### Author Contributions
- Li Yin conceived and designed the experiments, performed the experiments, analyzed the data, performed the computation work, prepared figures and/or tables, authored or reviewed drafts of the article, and approved the final draft.

### Data Availability
The code is available in the Supplementary File.

The raw data are available at Zenodo: UK-Union Dataset. (2022). UK-Union Dataset [Data set]. Zenodo. Available at https://doi.org/10.5281/zenodo.7131715.

### Supplemental Information
Supplemental information for this article can be found online at http://dx.doi.org/10.7717/peerj-cs.1668#supplemental-information.

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
