# Peer review of "Fuzzy information recognition and translation processing in English interpretation based on a generalized maximum likelihood ratio algorithm"

_PeerJ Computer Science, doi:10.7717/peerj-cs.1668_

## Round 0.1 · original submission · Major Revisions

Dear authors,

Thank you for your submission. Your article has not been recommended for publication in its current form. However, we do encourage you to address the concerns and criticisms of the reviewers and resubmit your article once you have updated it accordingly.

Best wishes,

**Language Note:** The review process has identified that the English language must be improved. PeerJ can provide language editing services - please contact us at [email protected] for pricing (be sure to provide your manuscript number and title). Alternatively, you should make your own arrangements to improve the language quality and provide details in your response letter. – PeerJ Staff

Reviewer 1 ·

Basic reporting

The proposed incorporation of fuzzy semantics within communication and writing processes seamlessly augments sentence flow and coherence, amplifying the conveyed content's expressive impact. I have carefully reviewed your paper and would like to provide you with some constructive feedback to improve the clarity and effectiveness of your research. Below, I have outlined specific suggestions for revision:

1. Conclude the abstract by summarizing the essential contributions the research makes to the field of interpretation and language processing.

2. Begin by clarifying the scope and significance of interpretation in the introduction.

3. When discussing vague language in oral communication, elaborate on its significance. How does it affect interpretation, and why is it important to address?

4. To provide a clearer context for this study, let's start by outlining the critical role interpretation plays in bridging language barriers.

5. Elaborate on the types of fuzzy information encountered during interpretation. Before diving into the algorithm, let's categorize and explore the various types of fuzzy information that interpreters commonly encounter.

Experimental design

6. Clarify the unpredictable nature of interpretation and its implications.

7. State the objective of the enhanced generalized maximum likelihood ratio algorithm (GLR) upfront.

8. Provide a clear link between the principles of natural language processing and interpretation.

9. Introduce the enhanced GLR technique as the main focus and explain the step-by-step approach of the fuzzy language processing method.

Validity of the findings

Comments are given in the basic reporting.

Additional comments

Comments are given in the basic reporting.

·

Basic reporting

This paper unveils an enhanced GLR methodology, tailored to address the intricacies of fuzzy information encountered in computer interpretation. In contrast to conventional techniques, the proposed approach attains heightened precision and adaptability. Computerized translation facilitates English learners in navigating basic dialogues, encompassing intricate verses and nebulous concepts alike.
In order for this study to be successfully accepted, please refine your manuscript according to the comments below


1. Define "fuzzy information" to avoid ambiguity. To address the challenges posed by fuzzy information, it is essential to establish a precise definition of this concept.

2. To better understand the complexity of interpretation, you must examine concrete examples of linguistic and cultural disparities.

3. To improve interpretation accuracy, you must delve into the intricate connections between language databases, vocabulary, grammar, and translation.

4. Highlight the practical importance of vague language in oral communication. Before presenting our algorithm, it is vital to underscore the practical significance of handling vague language effectively in oral communication.

5. Mention the details of your experimental setup, such as the dataset used, testing conditions, and evaluation metrics.

6. Ensure each paragraph flows logically into the next. To maintain a coherent structure, you must ensure that each section seamlessly leads into the following one.

7. In the final paragraph, offer some insights into what the experimental results indicate. How do they support the effectiveness of your proposed method?

8. Ensure that all sources are properly cited throughout the paper and that a list of references is included.

Experimental design

This paper unveils an enhanced GLR methodology, tailored to address the intricacies of fuzzy information encountered in computer interpretation. In contrast to conventional techniques, the proposed approach attains heightened precision and adaptability. Computerized translation facilitates English learners in navigating basic dialogues, encompassing intricate verses and nebulous concepts alike.
In order for this study to be successfully accepted, please refine your manuscript according to the comments below


1. Define "fuzzy information" to avoid ambiguity. To address the challenges posed by fuzzy information, it is essential to establish a precise definition of this concept.

2. To better understand the complexity of interpretation, you must examine concrete examples of linguistic and cultural disparities.

3. To improve interpretation accuracy, you must delve into the intricate connections between language databases, vocabulary, grammar, and translation.

4. Highlight the practical importance of vague language in oral communication. Before presenting our algorithm, it is vital to underscore the practical significance of handling vague language effectively in oral communication.

5. Mention the details of your experimental setup, such as the dataset used, testing conditions, and evaluation metrics.

6. Ensure each paragraph flows logically into the next. To maintain a coherent structure, you must ensure that each section seamlessly leads into the following one.

7. In the final paragraph, offer some insights into what the experimental results indicate. How do they support the effectiveness of your proposed method?

8. Ensure that all sources are properly cited throughout the paper and that a list of references is included.

Validity of the findings

This paper unveils an enhanced GLR methodology, tailored to address the intricacies of fuzzy information encountered in computer interpretation. In contrast to conventional techniques, the proposed approach attains heightened precision and adaptability. Computerized translation facilitates English learners in navigating basic dialogues, encompassing intricate verses and nebulous concepts alike.
In order for this study to be successfully accepted, please refine your manuscript according to the comments below


1. Define "fuzzy information" to avoid ambiguity. To address the challenges posed by fuzzy information, it is essential to establish a precise definition of this concept.

2. To better understand the complexity of interpretation, you must examine concrete examples of linguistic and cultural disparities.

3. To improve interpretation accuracy, you must delve into the intricate connections between language databases, vocabulary, grammar, and translation.

4. Highlight the practical importance of vague language in oral communication. Before presenting our algorithm, it is vital to underscore the practical significance of handling vague language effectively in oral communication.

5. Mention the details of your experimental setup, such as the dataset used, testing conditions, and evaluation metrics.

6. Ensure each paragraph flows logically into the next. To maintain a coherent structure, you must ensure that each section seamlessly leads into the following one.

7. In the final paragraph, offer some insights into what the experimental results indicate. How do they support the effectiveness of your proposed method?

8. Ensure that all sources are properly cited throughout the paper and that a list of references is included.

---

## Round 0.2 · accepted · Accept

Dear authors,

Thank you for the revision. The paper seems to be improved in the opinion of the reviewers. The paper is now ready to be published.

Best wishes,

Reviewer 1 ·

Basic reporting

All the changes are done.

Experimental design

All the changes are done.

Validity of the findings

All the changes are done.

Additional comments

No comment

·

Basic reporting

The manuscript is accepted for publication as all mentioned changes are well addressed.

Experimental design

The manuscript is accepted for publication as all mentioned changes are well addressed.

Validity of the findings

The manuscript is accepted for publication as all mentioned changes are well addressed.

Additional comments

The manuscript is accepted for publication as all mentioned changes are well addressed.